# Computed tomography with 6-year follow-up demonstrates the evolution of HTLV-1 related lung injuries: A cohort study

**Apio Ricardo Nazareth Dias**[1,2☯], **Waldonio de Brito Vieira**[1☯], **Valéria Marques Ferreira Normando**[2☯], **Karen Margarete Vieira da Silva Franco**[1☯], **Aline Semblano Carreira Falcão**[2], **Rita Catarina Medeiros de Sousa**[1], **Hellen Thais Fuzii**[1], **Luiz Fábio Magno Falcão**[2‡], **Juarez Antônio Simões Quaresma**[1,2‡]*

1 Federal University of Pará, Centre for Tropical Medicine, Belém, Pará, Brazil, 2 State University of Pará, Biological Sciences Health Centre, State University of Pará, Belém, Pará, Brazil

☯ These authors contributed equally to this work.
‡ LFMF and JASQ also contributed equally to this work.
* juarez@ufpa.br

**Data Availability Statement:** All relevant data are within the manuscript and its Supporting Information files.

## Abstract

Previous observational studies have demonstrated the development of pulmonary impairments in human T-lymphotropic virus type 1 (HTLV-1) infected individuals. The main observed lesions due to chronic inflammation of viral infection in situ are bronchiectasis and lung-scarring injuries. This lung inflammation may be the causal agent of restrictive and obstructive lung diseases, primarily in tropical spastic paraparesis/HTLV-1-associated myelopathy (TSP-HAM) patients. We conducted a prospective cohort study to compare spirometry and high-resolution computed tomography (HRCT) findings among 28 HTLV-1-carrier patients over the course of 6 years (2014–2019) (male/female: 7/21; mean age: 54.7 ± 9.5, range: 41–68 years). Chest HRCT exams revealed the development and evolution of lung lesions related to TSP-HAM: including centrilobular nodules, parenchymal bands, lung cysts, bronchiectasis, ground-glass opacity, mosaic attenuation, and pleural thickening. Spirometry exams showed maintenance of respiratory function, with few alterations in parameters suggestive of obstructive and restrictive disorders primarily in individuals with lung lesions and TSP-HAM. The findings of the present study indicate that pulmonary disease related to HTLV-1 is a progressive disease, with development of new lung lesions, mainly in individuals with TSP-HAM. To improve clinical management of these individuals, we recommend that individuals diagnosed with PET-MAH undergo pulmonary evaluation.

## Introduction

Human T-lymphotropic virus type 1 (HTLV-1) is a retrovirus and has a global infection incidence of ~ 20 million people, with a higher prevalence in Africa, Japan, and Latin America [1]. Brazil has a high prevalence, mainly in Maranhão, Bahia and Pará States, and is most common in the Brazilian Amazon rainforest regions [2, 3].

**Funding:** The author(s) received no specific funding for this work.

**Competing interests:** The authors have declared that no competing interests exist.

Studies have demonstrated the presence of lung lesions in people infected with HTLV-1. The major radiological findings are bronchiectasis, centrilobular nodules and ground glass opacities [4–6]; these lung injuries are attributable to chronic inflammation resulting from the effects of the virus in situ [7–10]. This lung inflammation is possibly the causal agent of obstruction of lung volume, flow limitation, and the development of restrictive and obstructive lung diseases, specifically, in tropical spastic paraparesis/HTLV-1-associated myelopathy (TSP-HAM) patients [7, 9, 11].

However, these studies and others have described only the high prevalence of lung injuries in patients infected by the virus [12–15], especially in patients who develop clinical inflammatory forms, such as TSP-HAM [7, 16, 17] and uveitis [18, 19].

Although recent publications, including a systematic review, have suggested a causal relation between HTLV-1 and the development of lung injury [10, 20], there are no studies which show the evolution of lung disease in HTLV-1 infected patients. This may be due to the lack of cohort studies needed to follow the evolution of these pulmonary symptoms in carrier patients or monitor infection progress in asymptomatic patients, making it impossible to establish a justifiable causal relationship between the virus and the emergence of lung injuries or symptoms. To the best of our knowledge, our study is the first to compare the findings of chest computed tomography (CT) and spirometry in a cohort of patients infected with HTLV-1 over a 6-year follow-up period; we demonstrate the clinical evolution of these patients and their lung injuries related to HTLV-1.

## Methods

### Ethics approval and consent to participate

This study was approved by the Research Ethics Committee involving human participants of the Evandro Chagas Institute (Approval Document No. 292.251), following the Declaration of Helsinki ethical precepts. All participants signed an informed consent form agreeing to participate in the study.

### Aim, study design, settings

This study aimed to compare the findings of chest computed tomography (CT) and spirometry in a cohort of patients infected with HTLV-1 in a 6-year follow-up period. This is a prospective cohort study at the Centre for Tropical Medicine Clinic, Belém, Pará, Brazil. All participants of the study had been recruited from a cohort of 48 individuals with HTLV-1, from both sexes, with and without TSP-HAM who were evaluated in 2014. They were referred to this research centre from a blood centre, primary healthcare unit, or after a relative was diagnosed with HTLV-1 infection. The diagnosis of each patient was confirmed by real-time polymerase chain reaction evaluation.

Patients' demographic and clinical characteristics were accessed from their medical records at assessment and again after 6 years. Patients with TSP-HAM were defined as those who presented with progressive spastic paraparesis without remission, some degree of gait changes perceptible by the patient, with or without sensory changes, and with or without urinary and anal sphincter signs or symptoms [21].

None of the patients had a history of congenital disease, tuberculosis, HIV infection, Pneumocystis Pneumonia, exposure to irritating chemicals, or childhood infections. None of the patients had documentation or clinical signs suggesting a diagnosis of pulmonary changes or respiratory disease before the HTLV-1 diagnosis. All patients were non-smokers.

The research participants (n = 28) were evaluated in 2014 and reassessed after a 6-year follow-up. Assessments included collection of clinical data such as diagnosis of TSP-HAM (De

Castro-Costa et al., 2006) [21], pulmonary symptomatology, previous lung diseases, chest CT and spirometry examinations. All procedures were repeated 6 years after the initial evaluation (Table 1).

## Participants, materials

The cohort, in this study (n = 48) comprised patients who were periodically monitored, of which 30 were classified as having TSP-HAM (19 women, aged 25–74 years, mean 52.45 ± 11.8 years, and 11 men, aged 38–64 years, mean 50.36 ± 8.6 years), and 18 as not having TSP-HAM (12 women, aged 39–66 years, mean 51± 9.7 years, and 6 men, aged 47–61 years, mean 52.3 ± 7.5 years). The cohort was evaluated, administered a spirometry exam and underwent CT in 2014, and was recruited for reassessment 6 years later, in 2019. Among these

**Table 1. Sample characteristics (2014–2019).**

| Sample Characteristics (n = 28) | | | |
|---|---|---|---|
| Men, n(%) | 7 (25) | | |
| Women, n(%) | 21 (75) | | |
| Age (mean± standard deviation) | 55.67±9.32 | | |
| **Economic condition** | | | |
| Less than 1 MW, n(%) | 2 (7.2) | | |
| 1 MW, n(%) | 14 (50) | | |
| 2 MW, n(%) | 5 (17.8) | | |
| 3 a 5 MW, n(%) | 7 (25) | | |
| **Schooling** | | | |
| Illiterate person, n(%) | 1 (3.6) | | |
| Elementary school, n(%) | 11 (39.2) | | |
| High School, n(%) | 14 (50) | | |
| Higher Education, n(%) | 2 (7.2) | | |
| **Clinical Findings** | **2014** | **2019** | **p-value** |
| Hypertension, n(%) | 6 (21.4) | 8 (28.6) | 0.75 |
| Cough, n(%) | 5 (17.8) | 5 (17.8) | 1 |
| Expectoration, n(%) | 3 (10.7) | 2 (7.2) | 1 |
| **Data collection** | **2014** | **2019** | **p-value** |
| **Clinical Form** | | | |
| TSP-HAM, n(%) | 18 (64.3) | 23 (82.1) | 0.00* |
| Assymptomatic carrier, n(%) | 10 (35.7) | 5 (17.9) | 0.00* |
| **CT Findings** | | | |
| Bronchiectasis, n(%) | 9 (32.1) | 12 (42.8) | 0.00* |
| Centrilobular nodules, n(%) | 7 (25) | 15 (53.6) | 0.00* |
| Ground glass opacities, n(%) | 4 (14.3) | 4 (14.3) | 1 |
| Mosaic Attenuation, n(%) | 1 (3.6) | 3 (10.7) | 0.10* |
| **Spirometry results** | | | |
| Normal, n(%) | 22 (78.6) | 16 (57.2) | 0.15 |
| Restriction, n(%) | 4 (14.3) | 3 (10.7) | 1 |
| Obstruction, n(%) | 2 (7.1) | 3 (10.7) | 1 |
| Isolated reduction of flows, n(%) | 0 | 6 (21.4) | 0.02* |

MW: Minimum wage. Normal: Normal spirometry. Restriction: suggestive of restrictive pulmonary dysfunction.
Obstruction: suggestive of obstructive pulmonary dysfunction.
* Fisher´s exact test (p<0.05).

patients, 20 were lost to tracking due to: death (n = 3), abandonment without justification (n = 1), moving from the city (n = 6), and change in the address and phone number (n = 10).

The remaining 28 patients were enrolled for reassessment. Twenty-three patients were classified as having TSP-HAM (17 women, aged 41–77 years, mean 55 ± 9.5 years, and 6 men, aged 43–64 years, mean 54.7 ± 9 years), and five (n = 5) classified as not having TSP-HAM (4 women, aged 42–68 years, mean 51.5 ± 12 years and 1 man, aged 51 years) (Fig 1).

The study was carried out at, the Ambulatory of the Tropical Medicine Centre in Belem, Pará. The HTLV-1 diagnosis was confirmed, and clinical evaluation, clinical classification, and spirometry exams of the study volunteers were performed. Chest CT scans were performed at the Diagnostic Imaging Service of the Santa Casa de Misericórdia do Pará Foundation, Brazil.

### Description of the processes

After agreeing to participate in the study, all individuals underwent two chest CT exams without intravenous contrast, at baseline and after 6 years, as follows. In the supine position, each patient took a deep breath and held it, inside the GE Multislice Brightspeed Edge Select CT scanner (GE Healthcare, Chalfont St Giles, UK) to obtain cross-sectional images of the chest with cuts of 1 mm collimation.

Each CT scan was analysed by two radiologists certified by the Brazilian College of Radiology and Diagnostic Imaging and with 6 and 9 years of experience, who did not know the patients' clinical diagnosis or the previous CT scan results. The kappa test was used to determine interobserver concordance ($k_1$ = 0.9298, $k_2$ = 0.9197). An iMac computer (Apple Inc., Cupertino, USA) and OsiriX MD software, version 5.8.5™ (Pixmeo Company, Bernex, Switzerland), were used to perform the pulmonary parenchyma analysis of each patient. All cuts above the diaphragm were evaluated and compared between assessment and reassessment.

The chest CT scan findings were classified in accordance with Fleischner Society: Glossary of Terms for Thoracic Imaging and Illustrated Brazilian Consensus of Terms and Fundamental Patterns in chest CT scans as: a) interlobular and septal thickening; b) bronchiectasis/bronchiolectasis: bronchial dilatation with respect to the accompanying pulmonary artery branch (signet ring sign), lack of tapering of the bronchi, and identification

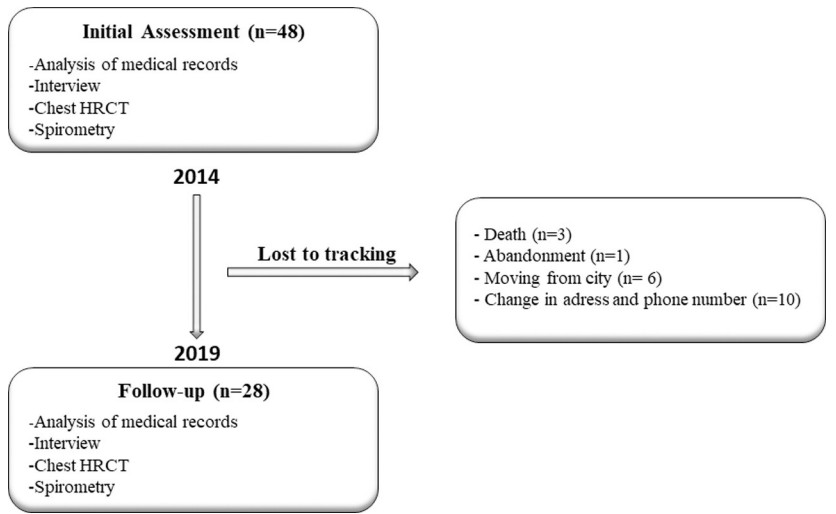

**Fig 1. Study flowchart (2014–2019).**

of bronchi within 1 cm of the pleural surface; c) parenchymal bands: an area of scarring in the lung parenchyma, reflecting pleuroparenchymal fibrosis (a parenchymal band is a linear opacity, usually 1–3 mm thick and up to 5 cm long that usually extends to the visceral pleura); d) pleural thickening; e) centrilobular nodules or small nodules with thickening of the wall or filling of the bronchiolar lumen; f) lung cyst or a rounded area in the lung parenchyma with a well-defined interface; g) ground-glass opacity, defined as increased density of the lung parenchyma that retains visible contours of the vessels and bronchi inside the affected area; and h) mosaic attenuation pattern or appearance of regions with distinct attenuation [22, 23].

All patients underwent two spirometry exams, at baseline and again after 6 years, as follows: the weight and height of individuals were verified. The exams were conducted with the individuals in the sitting position with a mouthpiece and nose clip. The device utilized was the Spirolab spirometer$^{TM}$ (Medical International Research, WI, USA). The vital capacity (VC) measurement was collected before forced vital capacity (FVC). For VC measurement, the individuals were instructed to completely fill and empty their lungs slowly. For FVC measurement, the individuals were instructed to inhale rapidly and completely in the mouthpiece and exhale with maximal force with a continued complete exhalation until the end of the test (6 seconds). These manoeuvres were conducted after demonstration of the correct technique by the examiner. The tests were repeated a minimum of three times and a maximum of eight times until three acceptable spirograms were obtained. The best obtained values were compared with the individual predicted values, following the guidelines established by the American Thoracic Society/ European Respiratory Society and Brazilian Consensus on Spirometry of Brazilian Society of Pneumology and Phthisiology for the interpretation of the results [24, 25].

The spirometric variables analysed were VC, FVC, forced expiratory volume in 1 second ($FEV_1$), the ratio of forced expiratory volume in 1 second to forced vital capacity ($FEV_1$/FVC), 25–75% forced expiratory flow ($FEF_{25-75\%}$), the ratio of the 25–75% forced expiratory flow to forced vital capacity ($FEF_{25-75\%}$/FVC), the maximum forced expiratory flow ($FEF_{max}$), 50% forced expiratory flow ($FEF_{50\%}$), 75% forced expiratory flow ($FEF_{75\%}$), and the maximum voluntary ventilation (MVV). The spirometry results of each patient were classified as normal, suggestive of restrictive dysfunction when VC obtained had values below the limit of normality [mild (60%), moderate (51%– 59%), severe (less than 50%)], or suggestive of obstructive dysfunction when FEV1/CVF obtained had values below the limit of normality [mild (60%), moderate (41%– 59%), severe (less than 40%)]. After analysis of the results for each variable, a comparison was made between the study groups in view of the values obtained in each period of the research.

## Statistical analysis

Data were analysed using GraphPad Prism software version 5.0™ (GraphPad Software, San Diego, USA). Shapiro wilk test was used to verify the normality of the sample. The sample size was calculated for assessment, considering the prevalence of abnormal pulmonary findings in 30.1% of patients with HTLV-1 infection [4] and 8.3% from a pilot study of 12 patients. Power of test was 80%, confidence Interval 95%, and alpha level 5%. The estimated sample was 28 individuals. Parametric data were assessed using the Paired T Test and analysis of variance. Non-parametric data were evaluated with the Wilcoxon signed rank-sum test. A 95% confidence interval was considered, and an alpha of 5% to refuse the null hypothesis, or a p value $\leq$ 0.05, was considered a statistically significant difference. The Kappa test was used to analyse the interobserver concordance.

## Results

### Chest computed tomography (CT) evaluation

The main findings in Chest CT analysis was the appearance of lung injury in (9/28) patients who did not presented lung injury at assessment, (Fig 2), and the intensification of existing lesions in (10/28) patients that showed parenchyma lesions at the initial evaluation (Figs 3–5). Only (6/28) patients showed no changes in chest CT findings.

At the sixth year, 9/12 G1 patients showed seven types of lung lesions, such as interlobular and septal thickening, pleural thickening, ground-glass opacity, bronchiectasis, centrilobular nodules, pulmonary cyst, and mosaic lung pattern. Moreover, there was an increase in the frequency of four types of lung lesions: ground-glass opacity, bronchiectasis, centrilobular nodules, and pleural thickening (Table 2).

### Spirometry (lung function) evaluation

The pulmonary function assessment of individuals in the cohort indicated that (4/28) had restriction, which after 6 years of monitoring, was (3/28). Individuals with obstruction (2/28) were also observed, and their number at re-evaluation increased (3/28). There were (6/28) individuals who, despite presenting normal values of pulmonary function, developed isolated reduction of flow to a mild degree. The spirometric variables of the G1 group are presented in Table 3 and the G2 group are presented in Table 4. The individuals i.e., HTLV-1 asymptomatic

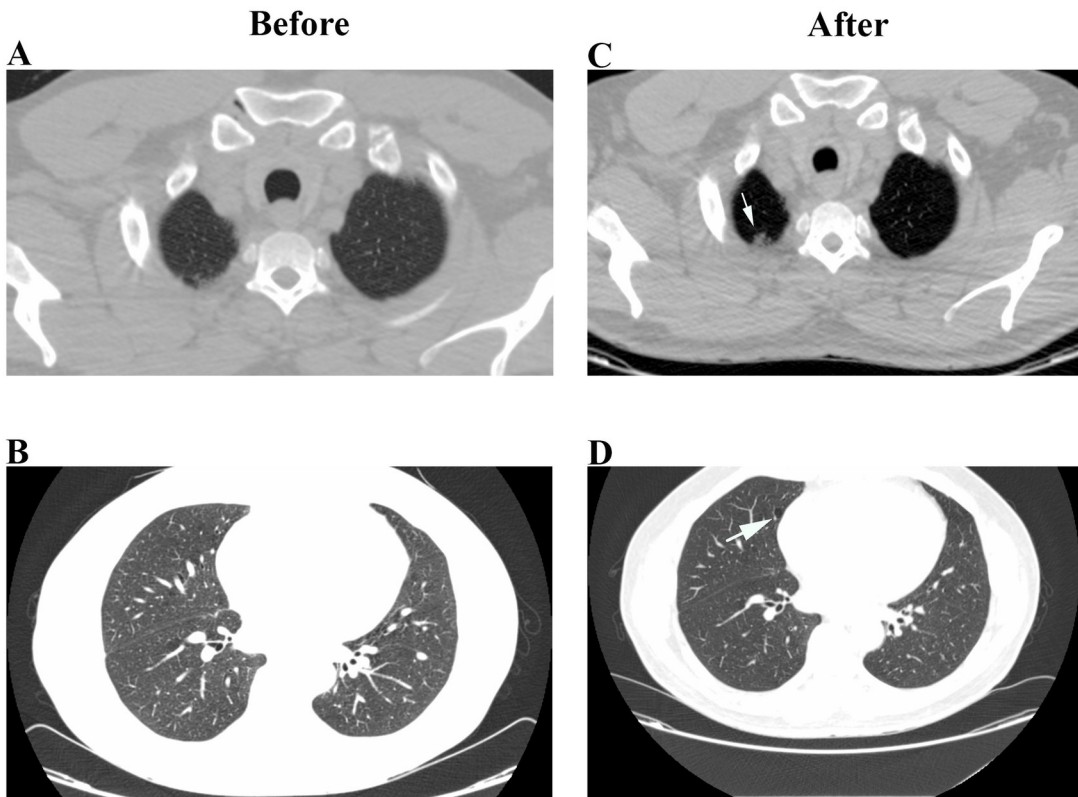

**Fig 2. Chest HRCT of a 52 year-old man (2014–2019).** (A) and (B) Before: No lesions; After (C): Centrilobular Nodules (Arrow); (D) After: Lung Cyst (Arrow).

**Before**

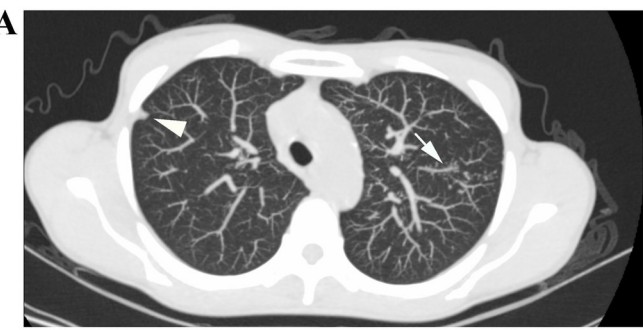

**After**

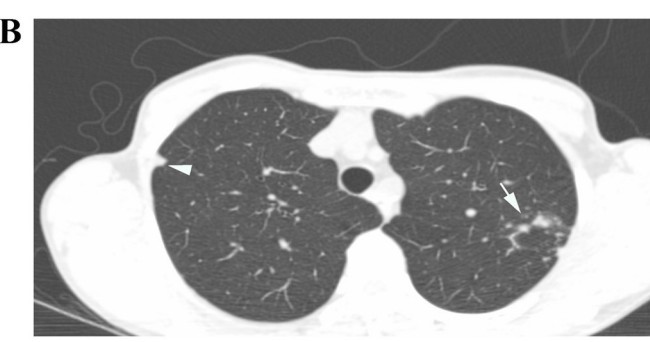

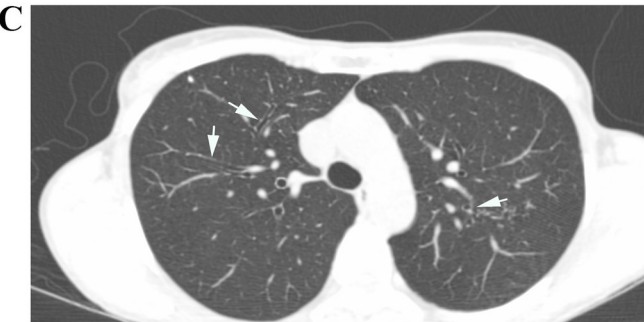

**Fig 3. Chest HRCT of a 47-year-old woman (2014–2019).** (A) Before: Pleural thickening (Arrowhead) and Centrilobular Nodules in a Tree in Bud shape (Arrow); (B) After: Evolution of Tree in Bud lesion (Arrow) and Pleural thickening (Arrowhead); (C) After: Bronchiectasis (Arrows).

carriers (n = 5; 4 women and 1 man) and individuals with TSP-HAM (n = 23; 17 women and 6 men) are presented in Table 5.

## Discussion

Cohort studies of this nature are difficult to perform and require a precise baseline assessment to ensure that participants do not have the condition of interest. The proportion of individuals who presented with new radiological findings of lung injury at follow-up is an important result of this study. In addition, we observed the evolution of pre-existing lung injuries, mainly those related to HTLV-1, according to scientific literature, and a few alterations of lung function, seemingly related to lung injury.

**Before**

**After**

Fig 4. Chest HRCT of a 64-year-old man (2014–2019). (A) Before: Bronchiectasis (Arrows), Centrilobular Nodules (Arrowheads); (B) After: Centrilobular Nodules in a Tree in Bud shape (Arrowheads); (C) After: Bronchiectasis (Arrows) and Centrilobular Nodules (Arrowheads); (D) After: Bronchiectasis (Arrows) and Pleural thickening (Arrowheads).

**Before**

**After**

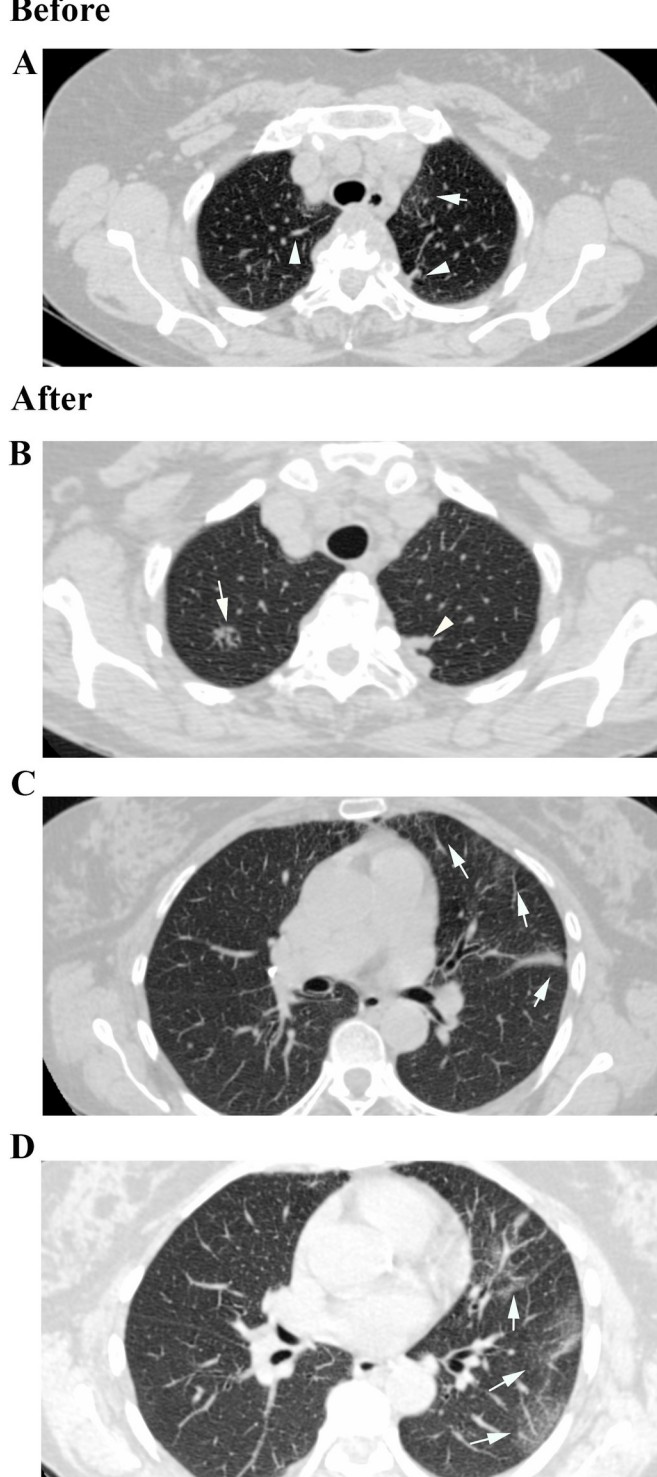

**Fig 5. Chest HRCT of a 45 year-old woman (2014–2019).** Woman, 45 years. (A) Before: Bronchiectasis (Arrows) and Centrilobular nodules (Arrowheads); (B) After: Centrilobular nodules (Arrow) and Pleural thickening (Arrowhead); (C) After: Ground-glass opacities (Arrows); (D) After: Ground-glass opacities (Arrows).

**Table 2. Chest CT findings at assessment and follow-up (2014–2019).**

| Individuals | Initial Classification | Initial Findings | Final Classification | Final Findings |
|---|---|---|---|---|
| #1 | Asy | none | Asy | CN and PB |
| #2 | HAM/TSP | none | HAM/TSP | CN |
| #3 | Asy | none | Asy | CN and LC |
| #4 | HAM/TSP | none | HAM/TSP | CN and PB |
| #5 | HAM/TSP | none | HAM/TSP | CN and BC |
| #6 | Asy | none | HAM/TSP | BC |
| #7 | HAM/TSP | none | HAM/TSP | none |
| #8 | Asy | none | HAM/TSP | CN |
| #9 | HAM/TSP | none | HAM/TSP | GG and MA |
| #10 | HAM/TSP | none | HAM/TSP | none |
| #11 | Asy | none | Asy | PT and BC |
| #12 | Asy | none | Asy | none |
| #13 | HAM/TSP | PB, BC | HAM/TSP | PB, BC |
| #14 | HAM/TSP | GG, BC,CN | HAM/TSP | GG, BC, CN |
| #15 | HAM/TSP | BC,CN,LC | HAM/TSP | BC,CN,LC |
| #16 | HAM/TSP | ST,GG,CN | HAM/TSP | ST,GG,CN |
| #17 | Asy | GG,BC | HAM/TSP | GG,BC |
| #18 | Asy | ST,BC | HAM/TSP | ST,BC |
| #19 | Asy | GG,PT | HAM/TSP | PB, BC, GG,PT and CN |
| #20 | HAM/TSP | PT, GG, CN,LC | HAM/TSP | GG and CN |
| #21 | HAM/TSP | ST | HAM/TSP | none |
| #22 | HAM/TSP | PB | HAM/TSP | CN |
| #23 | HAM/TSP | ST, PB, BC, CN | HAM/TSP | MAP |
| #24 | HAM/TSP | ST, GG, BC, CN, LC | HAM/TSP | GG, BC |
| #25 | HAM/TSP | ST, PB, PT, BC | HAM/TSP | BC, CN, MA |
| #26 | HAM/TSP | ST, PT, BC | HAM/TSP | PT, GG, BC, CN |
| #27 | HAM/TSP | ST, PT, BC, CN, MA | HAM/TSP | BC, CN |
| #28 | Asy | ST, BC, CN | Asy | ST, GG, BC |

Asy: Asymptomatic HTLV-1 carrier; HAM/TSP: HAM/TSP patients; ST: Interlobular septal thickening; PT: Pleural thickening; CN: Centrilobular nodules; PB: Parenchymal bands; BC: Bronchiectasis; GG: Ground-glass opacity; LC: Lung cyst; MA: Mosaic attenuation pattern.

The relationship between a higher pro-viral load (PVL) and lung lesion development has been demonstrated in previous studies, the PVLss were 16-fold higher in individuals with bronchiectasis or bronchiolitis [8], and these lesions were more frequent among TSP-HAM patients, as reflected in the CT findings of symptomatic patients [7]. In a recent community-based study, the rates of pulmonary disease were also very high among individuals without TSP-HAM who had high PVLs [26].

In TSP-HAM patients the increase of interferon-producing cells like CD4+, CD25+, CCR4 + and Foxp3- are correlated with increasing severity of inflammation [27–30]. The local immune and inflammatory responses play a key role in development of injuries that result in tissue damage [19, 31, 32]. Abnormal CT findings are seen in patients with HTLV-1 infection, especially those who develop TSP-HAM [7]. The screening of patients with HTLV-1 infection showed evolution in parenchymal lesions, with presence of chronic inflammation at the pulmonary level.

Bronchiectasis was the most characteristic CT finding in patients with HTLV-1-related lung disease [6]. Furthermore, injuries such as centrilobular nodules, ground-glass opacity,

**Table 3. Comparison of pulmonary lung function of G1 individuals without lung injury (n = 12) at chest CT (2014–2019).**

| Variable | G1 women (n = 11)** | | | | | |
|---|---|---|---|---|---|---|
| | Assess (%) | Reassess (%) | Pred | (a) | (b) | (c) |
| VC | 2.64 ± 0.36 (93) | 2.64 ± 0.53 (93) | 2.82± 0.27 | 0.05* | 0.21 | 0.97 |
| FVC | 2.72 ± 0.43 (96) | 2.50 ± 0.57 (88) | 2.82± 0.27 | 0.27 | 0.02* | 0.05* |
| FEV1 | 2.27 ± 0.36 (97) | 2.08 ± 0.47 (89) | 2.32± 0.24 | 0.52 | 0.02* | 0.09 |
| FEV1/FVC | 83.87 ± 3.18 (102) | 83.42 ± 7.04 (101) | 81.89 ± 1.30 | 0.08 | 0.26 | 0.83 |
| FEF 25–75% | 2.68 ± 0.59 (116) | 2.40 ± 0.80 (104) | 2.30 ± 0.24 | 0.03* | 0.65 | 0.30 |
| FEF 25–75%/FVC | 0.98 ± 0.18 (120) | 0.97 ± 0.29 (119) | 0.81 ± 0.03 | 0.00* | 0.10 | 0.86 |
| FEF Max | 5.66 ± 1.50 (86) | 5.63 ± 1.28 (85) | 6.55 ± 0.31 | 0.05* | 0.01* | 0.92 |
| FEF 50% | 3.53 ± 0.85 (117) | 3.25 ± 1.10 (108) | 3.00 ± 0.24 | 0.05* | 0.44 | 0.37 |
| FEF 75% | 1.02 ± 0.30 (122) | 0.89 ± 0.44 (107) | 0.83 ± 0.30 | 0.03# | 0.42 | 0.28 |
| MVV | 81.32 ± 18.52 (97) | 73.30 ± 16.94 (87) | 83.49 ± 27.23 | 0.37 | 0.09# | 0.09 |

values in (mean ± standard deviation). (%) percentage between measured and predicted values. (a) p-value between assessment and predicted values. (b) p-value between reassessment and predicted values. (c) p-value between assessment and reassessment values. Assess: Assessment. Reassess: Reassessment.

** between the 12 individuals, there was only a man, with spirometry values as follows (assess/reassess/pred): VC (3.64/4.12/4.05), FVC (3.87/3.92/4.05), FEV1 (3.18/3.00/3.38), FEV1/FVC (82.31/76.52/83.02), FEF 25–75% (3.22/2.43/3.56), FEF 25–75%/FVC (0.83/0.62/0.88), FEF máx (7.72/8.78/10.34), FEF 50% (4.05/3.36/4.49), FEF 75% (1.55/0.84/1.37), MVV (114.03/111.07/124.9). VC: Vital Capacity. FCV: Forced Vital Capacity. FEV1: Forced expiratory volume in one second. FEV1/FVC: ratio of forced expiratory volume in one second to forced vital capacity. FEF 25%-75%: 25%-75% forced expiratory flow. FEF 25%-75%/ FVC: ratio of 25%-75% forced expiratory flow to forced vital capacity. FEF max: maximum forced expiratory flow. FEF 50%: 50% forced expiratory flow. FEF 75%: 75% forced expiratory flow. MVV: maximum voluntary ventilation

*Paired T test (p<0.05).

#Wilcoxon (p< 0.05).

pleural thickening, and parenchymal bands were also found [5, 4, 7]. Previous observational studies suggest that these lung lesions are related to chronic inflammation at the pulmonary level due to HTLV-1 infection, leading to an increase in inflammatory cytokines in the

**Table 4. Comparison of pulmonary lung function of G2 individuals with lung injury (n = 16) at chest CT (2014–2019).**

| Variable | G2 women (n = 10) | | | | | | G2 men (n = 6) | | | | | |
|---|---|---|---|---|---|---|---|---|---|---|---|---|
| | Assess (%) | Reassess (%) | Pred | (a) | (b) | (c) | Assess (%) | Reassess (%) | Pred | (a) | (b) | (c) |
| VC | 2.56 ± 0.44 (85) | 2.64 ± 0.53 (87) | 3.01 ± 0.34 | 0.00* | 0.04* | 0.4 | 3.67 ± 1.05 (86) | 4.13 ± 1.12 (95) | 4.32 ± 0.13 | 0.2 | 0.6 | 0.09 |
| FVC | 2.58 ± 0.42 (85) | 2.50 ± 0.57 (83) | 3.01 ± 0.34 | 0.00* | 0.02* | 0.3 | 3.74 ± 1.27 (86) | 4.02 ± 1.08 (93) | 4.32 ± 0.13 | 0.3 | 0.5 | 0.4 |
| FEV1 | 2.06 ± 0.45 (83) | 1.99 ± 0.61 (80) | 2.47 ± 0.29 | 0.02* | 0.03* | 0.5 | 3.15 ± 0.96 (89) | 3.45 ± 0.88 (98) | 3.52 ± 0.16 | 0.4 | 0.8 | 0.2 |
| FEV1/FVC | 79.9 ± 11.68 (97) | 78.7 ± 9.93 (96) | 81.8 ± 2.12 | 0.58 | 0.29 | 0.6 | 84.9 ± 7.32 (104) | 84.6 ± 3.42 (104) | 81.3 ± 2.29 | 0.2 | 0.04* | 0.9 |
| FEF 25–75% | 2.17 ± 0.9 (89) | 2.11 ± 1.1 (86) | 2.43 ± 0.3 | 0.41 | 0.34 | 0.7 | 3.49 ± 1.0 (100) | 3.86 ± 0.7 (111) | 3.47 ± 0.4 | 0.9 | 0.37 | 0.2 |
| FEF 25–75%/FVC | 0.84 ± 0.39 (103) | 0.80 ± 0.35 (98) | 0.81 ± 0.06 | 0.80 | 0.88 | 0.7 | 0.96 ± 0.26 (120) | 1.01 ± 0.33 (126) | 0.80 ± 0.09 | 0.2 | 0.2 | 0.5 |
| FEF Max | 4.83 ± 1.21 (70) | 4.48 ± 1.53 (65) | 6.81 ± 0.43 | 0.00* | 0.00* | 0.4 | 6.76 ± 2.20 (63) | 6.92 ± 1.33 (65) | 10.6 ± 0.14 | 0.00* | 0.02* | 0.03* |
| FEF 50% | 2.76 ± 1.21 (87) | 2.79 ± 1.45 (88) | 3.15 ± 0.29 | 0.32 | 0.43 | 0.9 | 4.30 ± 1.23 (97) | 4.44 ± 0.71 (100) | 4.40 ± 0.42 | 0.8 | 0.9 | 0.6 |
| FEF 75% | 0.94 ± 0.44 (103) | 0.85 ± 0.53 (93) | 0.91 ± 0.17 | 0.80 | 0.70 | 0.3 | 1.76 ± 0.83 (133) | 1.98 ± 0.66 (150) | 1.32 ± 0.26 | 0.2 | 0.1 | 0.4 |
| MVV | 75.01 ± 17.4 (80) | 69.81 ± 19.7 (74) | 93.8 ± 12.9 | 0.01* | 0.00* | 0.2 | 111.2± 39.8 (84) | 121.9 ± 34.6 (92) | 132.1 ± 7.5 | 0.2 | 0.5 | 0.2 |

values in (mean ± standard deviation). (%) percentage between measured and predicted values. (a)p-value between assessment and predicted values. (b) p-value between reassessment and predicted values. (c) p-value between assessment and reassessment values. Assess: assessment. Reassess: reassessment. Pred: predict value. VC: Vital Capacity. FCV: Forced Vital Capacity. FEV1: Forced expiratory volume in one second. FEV1/FVC: ratio of forced expiratory volume in one second to forced vital capacity. FEF 25%-75%: 25%-75% forced expiratory flow. FEF 25%-75%/ FVC: ratio of 25%-75% forced expiratory flow to forced vital capacity. FEF max: maximum forced expiratory flow. FEF 50%: 50% forced expiratory flow. FEF 75%: 75% forced expiratory flow. MVV: maximum voluntary ventilation

*Paired T test (p<0.05).

**Table 5. Comparison of pulmonary lung function in individuals HTLV-1 asymptomatic carriers and TSP-HAM (2014–2019).**

| | Women | | | | | | | | | | | | |
| --- | --- | --- | --- | --- | --- | --- | --- | --- | --- | --- | --- | --- | --- |
| | HTLV-1 asymptomatic carriers (n = 4) | | | | | | TSP-HAM (n = 17) | | | | | | |
| Variable | Assess (%) | Reassess (%) | Pred | (a) | (b) | (c) | Assess (%) | Reassess (%) | Pred | (a) | (b) | (c) | (d) |
| VC | 2.73±0.29 (92) | 2.46±0.33 (83) | 2.97±0.33 | 0.16 | 0.00* | 0.03* | 2.57±0.41 (88) | 2.68±0.55 (92) | 2.90±0.32 | 0.00* | 0.09 | 0.25 | 0.45 |
| FVC | 2.79±0.55 (93) | 2.50±0.62 (84) | 2.97± 0.33 | 0.38 | 0.05* | 0.17 | 2.62±0.40 (90) | 2.50±0.56 (86) | 2.90±0.32 | 0.00* | 0.00* | 0.12 | 0.99 |
| FEV1 | 2.46±0.45 (100) | 2.13±0.49 (86) | 2.46±0.31 | 0.9 | 0.04* | 0.06 | 2.10±0.38 (88) | 2.02±0.54 (85) | 2.37±0.27 | 0.01* | 0.01* | 0.32 | 0.71 |
| FEV1/FVC | 88.66±3.42 (107) | 88.57±3.69 (107) | 82.56±1.54 | 0.03* | 0.03* | 0.9 | 80.41±8.50 (98) | 80.17±9.19 (98) | 81.73±1.73 | 0.79 | 0.47 | 0.91 | 0.03# |
| FEF 25–75% | 3.39±0.60 | 2.89±1.12 | 2.48±0.36 | 0.02* | 0.46 | 0.2 | 2.22±0.71 | 2.09±0.96 | 2.33±0.25 | 0.52 | 0.30 | 0.55 | 0.16 |
| FEF 25–75%/FVC | 1.22±0.19 | 1.16±0.34 | 0.83±0.03 | 0.02* | 0.14 | 0.5 | 0.84±0.27 | 0.82±0.32 | 0.80±0.05 | 0.56 | 0.81 | 0.78 | 0.07 |
| FEF Max | 6.66±1.46 | 6.45±1.49 | 6.72±0.36 | 0.9 | 0.6 | 0.8 | 4.94±1.22 | 4.76±1.31 | 6.67±0.40 | 0.00* | 0.00* | 0.55 | 0.03* |
| FEF 50% | 4.09±0.61 | 3.62±1.26 | 3.18±0.35 | 0.05* | 0.4 | 0.3 | 2.95±1.07 | 2.89±1.26 | 3.05±0.25 | 0.70 | 0.61 | 0.85 | 0.31 |
| FEF 75% | 1.46±0.43 | 1.05±0.47 | 0.94±0.21 | 0.02* | 0.6 | 0.1 | 0.87±0.25 | 0.83±0.48 | 0.85±0.14 | 0.52 | 0.85 | 0.29 | 0.43 |
| MVV | 92.80±22.29 | 77.60±17.14 | 91.06 ±11.57 | 0.8 | 0.03* | 0.12 | 74.91±15.49 | 70.24±18.37 | 93.10 ±12.49 | 0.00* | 0.00* | 0.16 | 0.47 |
| | Men** | | | | | | | | | | | | |
| | HTLV-1 asymptomatic carriers (n = 1) | | | | | | TSP-HAM (n = 6) | | | | | | |
| Variable | Assess (%) | Reassess (%) | Pred | (a) | (b) | (c) | Assess (%) | Reassess (%) | Pred | (a) | (b) | (c) | (d) |
| VC | 3.64 | 4.12 | 4.05 | - | - | - | 3.67±1.05 | 4.13±1.12 | 4.32±0.13 | 0.20 | 0.69 | 0.09 | - |
| FVC | 3.87 | 3.92 | 4.05 | - | - | - | 3.74±1.27 | 4.02±1.08 | 4.32±0.13 | 0.32 | 0.53 | 0.33 | - |
| FEV1 | 3.18 | 3 | 3.38 | - | - | - | 3.15±0.96 | 3.45±0.88 | 3.52±0.16 | 0.40 | 0.85 | 0.21 | - |
| FEV1/FVC | 82.31 | 76.52 | 83.02 | - | - | - | 84.97±7.32 | 87.04±4.90 | 81.32±2.29 | 0.29 | 0.04* | 0.31 | - |
| FEF 25–75% | 3.22 | 2.43 | 3.56 | - | - | - | 3.49±1.01 | 3.86±0.72 | 3.47±0.44 | 0.96 | 0.37 | 0.20 | - |
| FEF 25–75%/FVC | 0.83 | 0.62 | 0.88 | - | - | - | 0.96±0.26 | 1.01±0.33 | 0.80±0.09 | 0.21 | 0.23 | 0.50 | - |
| FEF Max | 7.72 | 8.78 | 10.34 | - | - | - | 6.76±2.20 | 6.92±1.33 | 10.63±0.14 | 0.00* | 0.02# | 0.91 | - |
| FEF 50% | 4.05 | 3.36 | 4.49 | - | - | - | 4.30±1.23 | 4.44±0.71 | 4.40±0.42 | 0.84 | 0.91 | 0.65 | - |
| FEF 75% | 1.55 | 0.84 | 1.37 | - | - | - | 1.76±0.83 | 1.98±0.66 | 1.32±0.26 | 0.28 | 0.11 | 0.46 | - |
| MVV | 114.03 | 111.07 | 124.9 | - | - | - | 111.24±39.86 | 122.62±33.78 | 132.10 ±7.53 | 0.28 | 0.53 | 0.18 | - |

values in (mean ± standard deviation). (%) percentage between measured and predicted values. (a)p-value between assessment and predicted values. (b) p-value between reassessment and predicted values. (c) p-value between assessment and reassessment values. (d) p-value between reassessment values of HTLV-1 asymptomatic carriers and TSP-HAM. Assess: assessment. Reassess: reassessment. Pred: predict value. VC: Vital Capacity. FCV: Forced Vital Capacity. FEV1: Forced expiratory volume in one second. FEV1/FVC: ratio of forced expiratory volume in one second to forced vital capacity. FEF 25%-75%: 25%-75% forced expiratory flow. FEF 25%-75%/ FVC: ratio of 25%-75% forced expiratory flow to forced vital capacity. FEF max: maximum forced expiratory flow. FEF 50%: 50% forced expiratory flow. FEF 75%: 75% forced expiratory flow. MVV: maximum voluntary ventilation.

*Paired T test (p<0.05).

#Wilcoxon (p<0.05).

**Between the Men there is only one individual HTLV-1 asymptomatic carrier.

bronchoalveolar fluid [18, 19, 32, 33] related to the development of pulmonary fibrosis through the activation of pulmonary fibroblasts [12].

Our study demonstrated that individuals with TSP-HAM clinical form have a high frequency of lung injury. It is possible that the mechanism of developing lesions is an in situ inflammatory process, diagnosed through bronchoalveolar lavage fluid analysis that has been shown in previous studies, which could be more evident among the TSP-HAM individuals [18, 19, 33, 34], with resultant induction of lung injuries, like bronchiectasis, fibrosis and scar lesions. Fibrosis itself could induce traction bronchiectasis in a cycle of chronic lung injury.

None of the patients had previous lung disease or environmental exposure associated with the occurrence and magnitude of their injuries during the follow-up period; thus, the development of TSP-HAM was considered to be the only associated factor. Although previous reports have described pulmonary involvement in individuals infected with HTLV-1 regardless of their clinical characteristics [12, 35, 36], most studies have found differences between patients with clinical features of inflammatory diseases related to asymptomatic carriers of viral infection [7, 32, 37, 38] with a higher frequency of pulmonary involvement among individuals with TSP-HAM.

Our spirometry analysis showed a decrease in lung function related to the lung injuries observed in the chest CT; this patient group showed a tendency of decline in VC, FVC, FEV1, $FEF_{25-75\%}$, and MVV values. Previous studies have shown that lung injury and altered lung function is more common in TSP-HAM individuals [7, 11], verifying the principal degree of lung involvement among those who developed the inflammatory form of the disease from HTLV-1.

The downward trend in VC, FVC, and FEV1, with the maintenance of normal FEV1/FVC values may indicate the development of restrictive lung disease; however the restriction must be confirmed by measuring lung values and documenting total lung capacity below normal limits [38]. The MVV measure is related to the level of physical activity in daily life, applied to individuals with chronic obstructive pulmonary disease [39], and the tendency of decreased values in HTLV-1 positive individuals may be related to the development of motor changes related to myelopathy associated with HTLV-1 infection or tropical spastic paraparesis [7]. The findings of this study suggest that changes in lung function occur mainly at the later stage of infection and depend on the degree of pulmonary parenchyma involvement.

This study shows essential yet previously unpublished data regarding the evolution of lung lesions related to HTLV-1, and is the first to perform a prospective 6-year follow-up analysis of the evolution of pulmonary involvement in these infected individuals conducted in the Brazilian Amazon where the virus is highly prevalent. This type of study design contributes to a better understanding of the natural history of the disease and its clinical evolution.

Despite the valuable findings of this study, it does not preclude limitations. First, due to the follow-up period, we ended up with a relatively small number of patients. The majority of HTLV-1 infected individuals remain asymptomatic, and only 3% develop TSP-HAM, adult T-cell leukaemia-lymphoma (ATL), or any clinical forms of HTLV-1 infection. The latency period of the disease is long, and thus, those people who did not develop clinical symptoms frequently did not return for periodic treatment.

Available prospective data for ATL and TSP-HAM are very limited, but existing observational studies have already demonstrated a relationship between lung injury and high viral loads, showing greater frequency in individuals with the clinical form of TSP-HAM. In this study we performed a follow-up using the gold standard method for evaluation of lung lesions, the chest HRCT.

Further cohort studies need to be conducted to determine the relationship between chronic inflammation and the development of lung injuries more accurately. Additionally, a greater number of individuals should be monitored for a prolonged period, to add to the clinical data and chest CT scans. Bronchoalveolar lavage fluid examination to determine the HTLV-1 proviral load, and determination of the level of inflammatory cytokines and chemokines and the pulmonary status, should also be performed.

The findings of the present study indicate that the pulmonary disease related to HTLV-1 is a progressive disease, with development of new lung lesions in individuals with TSP-HAM, diagnosis in whom is essentially clinical. To improve clinical management of these individuals, we recommend that individuals diagnosed with PET-MAH undergo pulmonary evaluation.

## Supporting information

**S1 File. Data from spirometry exams and chest CT analysis (2014–2019).**
(XLSX)

## Acknowledgments

We would like to thank the Federal University of Pará, the State University of Pará, and the Santa Casa de Misericórdia do Pará Foundation.

## Author Contributions

**Conceptualization:** Apio Ricardo Nazareth Dias, Luiz Fábio Magno Falcão, Juarez Antônio Simões Quaresma.

**Data curation:** Apio Ricardo Nazareth Dias, Luiz Fábio Magno Falcão, Juarez Antônio Simões Quaresma.

**Formal analysis:** Waldonio de Brito Vieira, Valéria Marques Ferreira Normando, Karen Margarete Vieira da Silva Franco, Aline Semblano Carreira Falcão, Luiz Fábio Magno Falcão.

**Investigation:** Apio Ricardo Nazareth Dias, Waldonio de Brito Vieira, Valéria Marques Ferreira Normando, Karen Margarete Vieira da Silva Franco, Rita Catarina Medeiros de Sousa, Hellen Thais Fuzii.

**Methodology:** Luiz Fábio Magno Falcão, Juarez Antônio Simões Quaresma.

**Project administration:** Juarez Antônio Simões Quaresma.

**Supervision:** Luiz Fábio Magno Falcão, Juarez Antônio Simões Quaresma.

**Validation:** Luiz Fábio Magno Falcão, Juarez Antônio Simões Quaresma.

**Visualization:** Aline Semblano Carreira Falcão, Luiz Fábio Magno Falcão, Juarez Antônio Simões Quaresma.

**Writing – original draft:** Apio Ricardo Nazareth Dias, Aline Semblano Carreira Falcão.

**Writing – review & editing:** Apio Ricardo Nazareth Dias, Rita Catarina Medeiros de Sousa, Hellen Thais Fuzii.

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
