## [Decision Letter · Decision Letter 0]

4 Nov 2021

PONE-D-21-26901Computed tomography with 6-year follow-up demonstrates the evolution of HTLV-1 related lung injuries: a cohort studyPLOS ONE

Dear Dr. Quaresma,

Thank you for submitting your manuscript to PLOS ONE. After careful consideration, we feel that it has merit but does not fully meet PLOS ONE’s publication criteria as it currently stands. Therefore, we invite you to submit a revised version of the manuscript that addresses the points raised during the review process.

We look forward to receiving your revised manuscript.

Kind regards,

Aleksandra Barac

Academic Editor

PLOS ONE

Journal Requirements:

Reviewers' comments:

Reviewer's Responses to Questions

**Comments to the Author**

1. Is the manuscript technically sound, and do the data support the conclusions?

Reviewer #1: Partly

Reviewer #2: No

Reviewer #3: Partly

2. Has the statistical analysis been performed appropriately and rigorously? 

Reviewer #1: Yes

Reviewer #2: N/A

Reviewer #3: Yes

3. Have the authors made all data underlying the findings in their manuscript fully available?

Reviewer #1: No

Reviewer #2: No

Reviewer #3: Yes

4. Is the manuscript presented in an intelligible fashion and written in standard English?

Reviewer #1: No

Reviewer #2: No

Reviewer #3: Yes

5. Review Comments to the Author

Reviewer #1: In the study “Computed tomography with 6-year follow-up demonstrates the evolution of HTLV-1 related lung injuries: a cohort study”, Quaresma et al. have presented alterations in pulmonary function and imaging features over time in HTLV-1 infected cases (particularly, in the TSP-HAM group). The main study topic is interesting and important to investigate. I have a few comments that are crucial to address before relying on the current study findings. Please read my comments below:

1. The authors have provided information regarding underlying pulmonary diseases or other infections such as TB, however, a major concern is the HIV co-infection. It is well established that HIV drastically increase the rate of HTLV-1 occurrence. On the other hand, HIV infection can lead to several other lung comorbidities (e.g., PCP infection). Please determine whether or not the study participants were diagnosed with HIV and if so, consider addressing HIV-related lung changes.

2. The current findings show that the isolate flow has shown significant difference in the reassessment from the baseline. As previously shown, aging causes a gradual decline in pulmonary function testing. The reported difference needs to be distinguished for the expected aging effect. For that, the authors need to correct for the age influence in their statistical models.

3. The authors have mentioned that student t-test has been used for part of the statistical analysis. The authors need to explicitly clarify what type of t-test has been used here as the longitudinal data are paired.

4. Please explain what you exactly mean by “urinary loss” under the clinical findings section of Table 1. Also, how does this relate to the other clinical features studied here?

5. Despite what has been mentioned in the discussion section, the current findings does not provide sufficient evidence that individuals with a TSP-HAM clinical form have a major probability of developing lung injury or if it is strongly related to a peculiar inflammatory process. Here, TSP_HAM patients were not compared to a control sample (in this case, non-TSP_HAM patients) to provide any statistical inference.

6. Please provide the full term for TSP-HAM at least for the first time mentioning it in the manuscript.

7. Finally, the article seems in need of major English improvement. I understand some of the practical difficulties in preparing the article, including of course the writing in English, but I believe this can be improved to further enhance its readability.

Reviewer #2: In the present study, the authors describe findings in 28 HTLV-1 carrier patients during a follow-up of 6 years. Their observations are based on findings in HRCT chest exams and spirometry examinations. They conclude that pulmonary disease related to HTLV-1 was a progressive disease with development of new lung lesions, mainly in individuals with TSP-HAM. Besides, they report that spirometry exams showed maintenance in the respiratory function with only few alterations in parameters, which suggested obstructive and respiratory disorders. The authors conclude by recommending HRCT to investigate lung lesions as soon as the investigation of TSP-HAM is confirmed.

In principle, the idea of conducting a longitudinal cohort study on HTLV-1 carrier patients is interesting. However, the collective of 28 patients is extremely small. This significantly reduces the relevance of the current study.

Abstract:

The authors should explain the abbreviations HTLV-1 and TSP-HAM. Also, the structure of the abstract should be improved.

Introduction:

- Very confusing. Some of the sentences do not make sense, e.g.: “Although recent publications, including a systematic review has suggested a causal relation between HTLV-1 and the development of lung injury [20,10], there are a inexistence of studies that shows the evolution of the lung disease in HTLV-1 infected patients, perhaps due to the lack of cohort studies needed to follow the evolution of these pulmonary symptoms in carrier patients or monitor infection progress in asymptomatic patients, making it impossible to establish a justifiable causal relationship between the virus and the emergence of lung injuries or symptoms.” Please revise grammar and spelling and elaborate..

- “Our study is the first to compare the findings of chest computed tomography (CT) and spirometry in a cohort of patients infected with HTLV-1 in a 6 years follow-up, we demonstrate the clinical evolution of these patients and their lung injuries related to HTLV-1.”

This assessment is misleading. There have been previous studies on HTLV-1 patients with HAM (e.g. PMID: 29281735). Only the 6-year-follow up is longer than in previous studies. This aspect should be clarified.

M&M:

- Why was there only one re-assessment after six years? Would have been interesting to have several re-assessments over the study interval.

- “Each CT scan was analysed by two radiologists who did not know the patient clinical diagnosis or the previous CT scan results.” – How experienced were the radiologists? Board-certified? This is very important in order to assess inter-observer agreement.

- Statistical analysis: Was a sample size calculation performed a priori? The number of 28 patients included appears too small to allow for adequate power..

Results:

- Structure has to be improved. Relevant findings do not become clear in its present form.

Discussion:

- Structure has to be improved. In its current form, it seems a bit thrown together. Organization should be enhanced. Instead of listing the individual results of the study, everything should be put into context.

- The conclusion “To a better clinical management of these individuals, Chest HRCT is recommended to investigate lung lesions as soon as the diagnosis of TSP-HAM is confirmed.” is not really supported by the data. Please elaborate.

Figures:

- Figure 1 needs to be redone. The design is very sloppy.

- Figure 2 is not convincing. This is a very non-specific finding.

General aspects:

Extensive English language revisions would be required to even consider publication. In its present form, the manuscript is not suitable for publication.

Reviewer #3: This study demonstrate the lung injuries and pulmonary function evolutions of this individuals through a follow-up of six years. They conducted a prospective cohort study to compare spirometry and high-resolution computed tomography (HRCT) findings among 28 HTLV-1-carrier patients during 6 years (2014–2019) of follow-up (male/female: 7/21; mean age: 54.7 ±

33 9.5, range: 41–68 years). The results were interesting. Therefore, I recommend it for publication after checking the manuscript grammatically

6. PLOS authors have the option to publish the peer review history of their article (what does this mean?). If published, this will include your full peer review and any attached files.

Reviewer #1: No

Reviewer #2: No

Reviewer #3: **Yes: **Amr A. Abd-Elghany

---

## [Author Response · Author response to Decision Letter 0]

26 Nov 2021

Reviewer #1: In the study “Computed tomography with 6-year follow-up demonstrates the evolution of HTLV-1 related lung injuries: a cohort study”, Quaresma et al. have presented alterations in pulmonary function and imaging features over time in HTLV-1 infected cases (particularly, in the TSP-HAM group). The main study topic is interesting and important to investigate. I have a few comments that are crucial to address before relying on the current study findings. Please read my comments below:

1. The authors have provided information regarding underlying pulmonary diseases or other infections such as TB, however, a major concern is the HIV co-infection. It is well established that HIV drastically increase the rate of HTLV-1 occurrence. On the other hand, HIV infection can lead to several other lung comorbidities (e.g., PCP infection). Please determine whether or not the study participants were diagnosed with HIV and if so, consider addressing HIV-related lung changes.

Answer: Thank you for your comments. We agree. None of the participants has previous history of co-infection, including HIV and PCP infection. This information was included on manuscript text (lines 94-95).

2. The current findings show that the isolate flow has shown significant difference in the reassessment from the baseline. As previously shown, aging causes a gradual decline in pulmonary function testing. The reported difference needs to be distinguished for the expected aging effect. For that, the authors need to correct for the age influence in their statistical models.

Answer: Thank you for your comments. The follow-up time was very short, not being enough for age influence the sample. For prevent that, the comparisons included the predict values of spirometry that was related to the individuals age. 

3. The authors have mentioned that student t-test has been used for part of the statistical analysis. The authors need to explicitly clarify what type of t-test has been used here as the longitudinal data are paired.

Answer: We agree. The Paired T test were used. This information was included in the Statistical Analysis section and in the Table legends.

4. Please explain what you exactly mean by “urinary loss” under the clinical findings section of Table 1. Also, how does this relate to the other clinical features studied here?

Answer: We agree. The urinary loss was a symptom included in the research protocol because their association with TSP-HAM. By the way, it is not related with the other clinical features, in special the lung lesions, studied here. We excluded this information from the table.

5. Despite what has been mentioned in the discussion section, the current findings does not provide sufficient evidence that individuals with a TSP-HAM clinical form have a major probability of developing lung injury or if it is strongly related to a peculiar inflammatory process. Here, TSP_HAM patients were not compared to a control sample (in this case, non-TSP_HAM patients) to provide any statistical inference.

Answer: We agree. The manuscript text was modified to indicate that we find a high frequency of lung injury among TSP-HAM individuals (lines 336-339).

6. Please provide the full term for TSP-HAM at least for the first time mentioning it in the manuscript.

Answer: We agree. The full term was provided in the manuscript text (line 56) and abstract (line 29).

7. Finally, the article seems in need of major English improvement. I understand some of the practical difficulties in preparing the article, including of course the writing in English, but I believe this can be improved to further enhance its readability.

Answer: We agree. The English of the manuscript was improved, the modifications were marked along the text.

Reviewer #2: In the present study, the authors describe findings in 28 HTLV-1 carrier patients during a follow-up of 6 years. Their observations are based on findings in HRCT chest exams and spirometry examinations. They conclude that pulmonary disease related to HTLV-1 was a progressive disease with development of new lung lesions, mainly in individuals with TSP-HAM. Besides, they report that spirometry exams showed maintenance in the respiratory function with only few alterations in parameters, which suggested obstructive and respiratory disorders. The authors conclude by recommending HRCT to investigate lung lesions as soon as the investigation of TSP-HAM is confirmed.

In principle, the idea of conducting a longitudinal cohort study on HTLV-1 carrier patients is interesting. However, the collective of 28 patients is extremely small. This significantly reduces the relevance of the current study.

Abstract:

The authors should explain the abbreviations HTLV-1 and TSP-HAM. Also, the structure of the abstract should be improved.

Answer: We agree. The abbreviations was explained (line 26 and 29) at the abstract and its structure was improved.

Introduction:

- Very confusing. Some of the sentences do not make sense, e.g.: “Although recent publications, including a systematic review has suggested a causal relation between HTLV-1 and the development of lung injury [20,10], there are a inexistence of studies that shows the evolution of the lung disease in HTLV-1 infected patients, perhaps due to the lack of cohort studies needed to follow the evolution of these pulmonary symptoms in carrier patients or monitor infection progress in asymptomatic patients, making it impossible to establish a justifiable causal relationship between the virus and the emergence of lung injuries or symptoms.” Please revise grammar and spelling and elaborate..

Answer: We agree. The English grammar and spelling were revised, the text between lines 61-70 were modified for better comprehension.

- “Our study is the first to compare the findings of chest computed tomography (CT) and spirometry in a cohort of patients infected with HTLV-1 in a 6 years follow-up, we demonstrate the clinical evolution of these patients and their lung injuries related to HTLV-1.”

This assessment is misleading. There have been previous studies on HTLV-1 patients with HAM (e.g. PMID: 29281735). Only the 6-year-follow up is longer than in previous studies. This aspect should be clarified.

Answer: We agree. The manuscript text was modified for better comprehension (lines 67, 69 and 70).

M&M:

- Why was there only one re-assessment after six years? Would have been interesting to have several re-assessments over the study interval.

Answer: We agree. Centre for Tropical Medicine is a research institution that conduces clinical follow-up of these patients, however, there is a lack of adequate re-assessments with complementary exams (e.g. Spirometry, or Chest HRCT), this data collection only was possible with the development of research projects from CTM´s Graduate Program students. The follow-up time over six years refers to the time lapse between the initial study carried out in this cohort, a cross-sectional study, and this follow-up, both carried out by the present researchers.

- “Each CT scan was analysed by two radiologists who did not know the patient clinical diagnosis or the previous CT scan results.” – How experienced were the radiologists? Board-certified? This is very important in order to assess inter-observer agreement.

Answer: We agree. The informations about how experienced were the radiologists and their board certification was provided in the manuscript text. (lines 135-136)

- Statistical analysis: Was a sample size calculation performed a priori? The number of 28 patients included appears too small to allow for adequate power..

Answer: We agree. The sample size was calculated for the initial assessment. The informations was included at manuscript text (lines 184-188).

Results:

- Structure has to be improved. Relevant findings do not become clear in its present form.

Answer: We agree. The structure was improved in results do become clear the relevant findings of our study. Lines (197-200) and lines (246-248)

Discussion:

- Structure has to be improved. In its current form, it seems a bit thrown together. Organization should be enhanced. Instead of listing the individual results of the study, everything should be put into context.

Answer: We agree. The discussion was restructured for better comprehension (lines 323-332).

- The conclusion “To a better clinical management of these individuals, Chest HRCT is recommended to investigate lung lesions as soon as the diagnosis of TSP-HAM is confirmed.” is not really supported by the data. Please elaborate.

Answer: The conclusion was modified for better comprehension and in accordance with the research results. We recommend the pulmonary evaluation of HAM-TSP individuals (lines 387-388).

Figures:

- Figure 1 needs to be redone. The design is very sloppy.

Answer: We agree. The figure was redone.

- Figure 2 is not convincing. This is a very non-specific finding.

Answer: Thank you for your comments. We agree. Scientific literature do not show typical findings of lung lesions related to HTLV-1 infection, but our studies developed in Eastern Amazon, has shown this lung lesion, centrilobular nodule, as one of the main findings among individuals.

General aspects:

Extensive English language revisions would be required to even consider publication. In its present form, the manuscript is not suitable for publication.

Answer: We agree. The manuscript language was revised as solicited.

Reviewer #3: This study demonstrate the lung injuries and pulmonary function evolutions of this individuals through a follow-up of six years. They conducted a prospective cohort study to compare spirometry and high-resolution computed tomography (HRCT) findings among 28 HTLV-1-carrier patients during 6 years (2014–2019) of follow-up (male/female: 7/21; mean age: 54.7 ±

33 9.5, range: 41–68 years). The results were interesting. Therefore, I recommend it for publication after checking the manuscript grammatically

Answer: We agree. The English was revised.

---

## [Editor Report · Decision Letter 1]

13 Dec 2021

Computed tomography with 6-year follow-up demonstrates the evolution of HTLV-1 related lung injuries: a cohort study

PONE-D-21-26901R1

Dear Dr. Quaresma,

We’re pleased to inform you that your manuscript has been judged scientifically suitable for publication and will be formally accepted for publication once it meets all outstanding technical requirements.

Kind regards,

Aleksandra Barac

Academic Editor

PLOS ONE

---

## [Editor Report · Acceptance letter]

19 Dec 2021

PONE-D-21-26901R1 

Computed tomography with 6-year follow-up demonstrates the evolution of HTLV-1 related lung injuries: A cohort study 

Dear Dr. Quaresma:

I'm pleased to inform you that your manuscript has been deemed suitable for publication in PLOS ONE. Congratulations! Your manuscript is now with our production department. 

Kind regards, 

on behalf of

Dr. Aleksandra Barac 

Academic Editor

PLOS ONE